

# Analysis of the factors influencing the proximity and agreement between critical power and maximal lactate steady state: a systematic review and meta-analyses

Lorenzo Micheli[1,*], Francesco Lucertini[1,*], Tommaso Grossi[1], Silvia Pogliaghi[2,3,4], Daniel A. Keir[4,5] and Carlo Ferri Marini[1,6]

[1] Department of Biomolecular Sciences – Division of Exercise and Health Sciences, University of Urbino Carlo Bo, Urbino, Marche, Italy
[2] Department of Neurosciences, Biomedicine and Movement Sciences, University of Verona, Verona, Veneto, Italy
[3] Canadian Center for Activity and Ageing, The University of Western Ontario, London, Ontario, Canada
[4] School of Kinesiology, The University of Western Ontario, London, Ontario, Canada
[5] Toronto General Hospital Research Institute, Toronto General Hospital, Toronto, Ontario, Canada
[6] Department of Human Movement Sciences, University of Groningen, University Medical Center Groningen, Groningen, Netherlands
[*] These authors contributed equally to this work.

Corresponding author
Carlo Ferri Marini,
c.ferri.marini@umcg.nl

## ABSTRACT

Identifying the boundary between heavy and severe exercise domains is crucial since it demarcates the transition from sustainable to unsustainable exercise. This systematic review aimed to determine differences and agreement between two indices used to determine this boundary, namely critical power (CP) and maximal lactate steady state (MLSS), and how moderators may affect these differences. Ten out of 782 studies found were included in the meta analyses. Random effect meta-analyses were performed to evaluate the mean differences (MD) between CP and MLSS, and moderators' effect on MD was assessed using meta-regression. CP and MLSS agreement was tested using Bland-Altman meta-analyses on the limits of agreements (LoA) of the MD. Power output (PO) at CP was higher (MD (95% LoA) = 12.42 [−19.23; 44.08] W, $p = 0.005$) than PO at MLSS, with no differences between CP and MLSS in terms of oxygen uptake (MD (95% LoA) = 0.09 [−0.34; 0.52] L·min$^{-1}$, $p = 0.097$), heart rate (MD (95% LoA) = 0.61 [−15.84; 17.05] bpm, $p = 0.784$), and blood lactate concentration (MD (95% LoA) = 1.63 [−2.85; 6.11] mM, $p = 0.240$). Intensities at CP ($p = 0.002$) and MLSS ($p = 0.010$) influenced the MD expressed in W. In conclusion, solely when expressed in PO, CP was higher than MLSS, with larger differences in fitter and younger individuals, emphasizing the possible effect of the indicators used for assessing exercise intensity. Finally, the high interindividual variability observed in the differences between CP and MLSS could compromise their interchangeability in predicting the heavy to severe boundary regardless of the parameter used to assess exercise intensity.

## INTRODUCTION

In exercise physiology, it is widely accepted that there are distinct aerobic exercise intensity domains. These domains, namely moderate, heavy, and severe, differ on whether and how long it takes for oxygen uptake ($\dot{V}O_2$) and blood lactate concentration (BLC) to attain a stable, submaximal level, and how long exercise can be sustained before exhaustion. Within the moderate exercise domain (*i.e.,* intensities below lactate threshold (LT) or gas exchange threshold (GET)), it takes approximately 2–3 min for $\dot{V}O_2$ to reach a steady state, and BLC remains at resting levels (*Carter et al., 2002*). During heavy-intensity exercise (*i.e.,* between LT or GET and respiratory compensation point (RCP)), which is characterized by the presence of a secondary $\dot{V}O_2$ slow component, it takes about 15–20 min for $\dot{V}O_2$ to stabilize, while BLC stabilizes at an elevated level (*Caputo & Denadai, 2004*). Conversely, in the severe intensity domain achieving a metabolic steady state in both $\dot{V}O_2$ and BLC is no longer possible, with $\dot{V}O_2$ tending to reach the maximal oxygen uptake ($\dot{V}O_{2max}$) (*Hill, Poole & Smith, 2002*).

The metabolic stress generated within each intensity domain is unique but tends to be similar between individuals, even if they differ in fitness levels (*Iannetta et al., 2020*). Indeed, it is widely accepted that specific boundaries demarcating exercise intensity domains do exist. Although there is a consensus on using the LT, or its estimation based on gas exchange (GET) or ventilatory data (the first ventilatory threshold), to determine the boundary between moderate and heavy intensity domains (*Poole et al., 2021*), the determination of the best index for representing the boundary between heavy and severe intensity domains remains a topic of debate (*Jones et al., 2019*; *Keir et al., 2015*; *Keir, Pogliaghi & Murias, 2018*; *Maturana et al., 2016*; *Poole et al., 2021*). Clarifying the optimal methods for determining this threshold, commonly referred to as maximal metabolic steady state (MMSS) (*Jones et al., 2008*), is critical as it marks the metabolic boundary between the exercise sustainable and unsustainable in a homeostatic condition (*Keir et al., 2015*; *Rossiter, 2011*). Various conceptual models and terms have emerged to define this boundary in exercise physiology (*Poole et al., 2021*). Among the approaches proposed in the literature, the maximal lactate steady state (MLSS) (*Beneke, 1995*; *Beneke, 2003*) and the critical power (CP) are the most widely used (*Jones et al., 2019*; *Poole et al., 2021*). The MLSS refers to the highest exercise intensity sustainable without continuous blood lactate accumulation (*Beneke, 1995*; *Heck et al., 1985*). Alternatively, the concept of CP is considered a valuable tool for understanding the metabolic responses during exercise (*Poole et al., 2016*) and has been recently proposed as the gold standard for representing the transition from heavy to severe boundary (*Jones et al., 2019*).

It has been suggested that both these concepts could be considered similar (*Poole et al., 2016*; *Rossiter, 2011*; *Svedahl & MacIntosh, 2003*) and can be used to estimate the MMSS (*Jones et al., 2019*), previous studies have found a variable relationship between the power outputs at which they occur (*Dekerle et al., 2003*; *Greco et al., 2012*; *Keir et al., 2015*; *Maturana et al., 2016*; *Pringle & Jones, 2002*). These variations seem to be attributable to differences in the methods employed for their determination (*Iannetta et al., 2022*; *Borszcz et al., 2024*). Indeed, various methodological factors highly influence MLSS and CP

(*Iannetta et al., 2022*; *Dotan, 2022*). For instance, when determining MLSS, which involves performing multiple (*i.e.,* 3 to 5) constant power exercises lasting 30 min each, several factors can impact the identification of MLSS (*Iannetta et al., 2022*). In particular, the time interval chosen for identifying lactate steady state (*i.e.,* less than one mmol (mM) increase between 10th and 30th or between 20th and 30th min of exercise) and the power output intervals selected between trials (*i.e.,* ±10 or 15 W) have been proven to affect MLSS estimation (*Beneke, 2003*; *Heck et al., 1985*; *Iannetta et al., 2022*; *Svedahl & MacIntosh, 2003*; *Nixon et al., 2021*).

Similarly, CP estimation, which derives from the power-time relationship established through four to five bouts of exhaustive exercise, can yield different results depending on the number and duration (*i.e.,* time to exhaustion (TTE)) of exhaustion trials conducted, the models employed for its estimation (*Bishop, Jenkins & Howard, 1998*; *Maturana et al., 2018*), and pedal cadence (*Dotan, 2022*). In this regard, a recent meta-analysis by *Galan-Rioja et al. (2020)* highlighted substantial differences (of approximately 30 W) when comparing MLSS and CP, which could be attributed to the multiple factors discussed earlier.

Although the meta-analysis of *Galan-Rioja et al. (2020)* aimed at comparing heavy-to severe-exercise boundaries, several important aspects of the association between these boundaries were unaddressed. First, the meta-analysis (*Galan-Rioja et al., 2020*) focused solely on correlation coefficients between CP and MLSS and did not consider their mean difference (MD) nor analyzed their agreement using a meta-analytic approach. Second, the meta-analysis (*Galan-Rioja et al., 2020*) did not assess the possible effect of moderators that can influence the relation between CP and MLSS, such as the PO corresponding to MLSS and CP, as well as the age of participants. These limitations were highlighted by a recent systematic review (*Borszcz et al., 2024*), which provided a comprehensive explanation of the agreement between CP and MLSS in different types of dynamic exercise (*e.g.,* running, cycling, rowing, and swimming). Third, the association between MLSS and CP was analyzed only in terms of PO and the effect of using different variables (*e.g.,* $\dot{V}O_2$, heart rate (HR), BLC, PO) to assess exercise intensity on the association between CP and MLSS during prolonged exercise was not evaluated (*Borszcz et al., 2024*; *Galan-Rioja et al., 2020*). This third aspect is of paramount importance for both researchers and practitioners because of the dissociation that may occur between the variables used to assess exercise intensity due to the different time-depended adjustments (*e.g.,* cardiovascular drift and slow component) (*Ferri Marini et al., 2024*; *Cunha et al., 2011*; *Teso, Colosio & Pogliaghi, 2022*; *Ferri Marini et al., 2022a*). In this regard, in order to accurately monitor training responses and optimize exercise prescription, it is crucial to identify which physiological parameter (*i.e.,* $\dot{V}O_2$, HR, BLC) is associated with a smaller difference between CP and MLSS.

Therefore, the primary aim of this study was to assess the association and the agreement between CP and MLSS using a meta-analytical approach. The secondary aim was to assess how different exercise intensity indicators (*i.e.,* PO, $\dot{V}O_2$, HR, and BLC) and moderators (*i.e.,* PO at both CP and MLSS, and age of participants) affect the proximity between CP and MLSS.

**Table 1  PICOS principle of inclusion criteria.**

| Parameters | Inclusion criteria |
|---|---|
| Population | Apparently healthy adults between 18 and 65 years old |
| Intervention | Contant intensity exercise at CP and MLSS |
| Comparison | CP and MLSS |
| Outcome | PO, $\dot{V}O_2$, BLC, or HR |
| Study design | Crossover or within subject design |

Notes.
CP, critical power; MLSS, maximal lactate steady state; PO, power output; $\dot{V}O_2$, oxygen uptake; BLC, blood lactate concentration; HR, heart rate.

# MATERIALS AND METHODS

## Protocol and registration

The present systematic review was registered with the International Prospective Register of Systematic Reviews (PROSPERO; Registration number CRD42021261155 Available from: https://www.crd.york.ac.uk/prospero/display_record.php?ID=CRD42021261155). Additionally, the present systematic review was conducted in accordance with the Preferred Reporting Items for Systematic Reviews and Meta-Analyses (PRISMA) statement for reporting systematic reviews and meta-analyses (*Page et al., 2021*).

## Search strategy

Two independent reviewers conducted a literature search on the following databases: PubMed, Web of Science, and Scopus.

Title, abstract, and keyword search fields were searched using the following keywords: critical power, maximal metabolic steady state, maximal lactate steady state, respiratory compensation point, and thresholds. The above keywords were searched using different combinations created through the utilization of the Boolean operators (*i.e.,* AND and OR) and the search criteria are available as Supplemental Information (see SDC1_Keyword search strategy). Additionally, weekly alerts were set up on Scopus, Web of Science, and PubMed, and the papers published until March 2023 were screened according to the inclusion criteria listed below.

Two reviewers (LM and CFM) independently performed the studies' identification, screening, eligibility, and inclusions. If any disagreement was present it was resolved by the senior author (FL) after an open discussion.

## Inclusion and exclusion criteria

Inclusion criteria were set according to PICOS (population, intervention, comparison, outcome, and study design) principle as shown in Table 1. This meta-analysis included studies with participants aged from 18 to 65 years published in English and Portuguese languages. The inclusion of articles was not established a priori. This is because Portuguese was the only language, aside from English, in which articles meeting the established inclusion criteria were available.

Data were extracted only from the studies that performed a constant intensity trial at CP and MLSS exercise intensities. The inclusion of studies that had performed constant trials

at CP and MLSS was critical since physiological indicators (*e.g.*, $\dot{V}O_2$) related to the two thresholds (*i.e.,* CP and MLSS) were directly measured during prolonged exercises and not extrapolated, for instance, from the incremental test. Articles were excluded if they were duplicates and did not meet the above-mentioned inclusion criteria.

## Data extraction

The studies included were used to extract the relevant exercise intensity indicators and other pertinent variables. If the original study did not report the necessary information for the following analyses, the authors were contacted for clarifications on the missing results or additional methodological information.

## Statistical analyses

All statistical analyses were performed with R software (*R Core Team, 2021*) using metafor (version 3.8.1), dmetar (version 0.0.9000), clubSandwich (version 0.5.8), and forester (version 0.2.0) packages (*Pustejovsky, 2022*; *Viechtbauer, 2010*; *Harrer et al., 2019*), using $\alpha$ equal to 0.05.

The following statistical analyses, aimed at assessing the differences and agreement between CP and MLSS, were performed separately for each measurement unit (*i.e.,* PO, $\dot{V}O_2$, HR, BLC) used to assess exercise intensity, due to their different scales.

Since CP and MLSS exercise intensities were correlated data deriving from the same individual, the mean difference (MD) between CP and MLSS and their standard deviation (SD) were used to compute the effect size used in the following analyses to account for paired measures and the intra-individual correlation between the two exercise intensities (*Gibbons, Hedeker & Davis, 1993*; *Higgins et al., 2003*). For each study, the MD and their SD were retrieved from the original studies or imputed from the CP and MLSS means, SDs, and Pearson correlation ($r$) or MD and limits of agreements (LoA) of the selected studies, as proposed by *Ferri Marini et al. (2022b)*. When the information needed to compute the SD of the MD were not present nor provided by the authors upon request, the median $r$ between CP and MLSS of the studies with the same measurement unit was used to compute the SD of the MD (*Ferri Marini et al., 2022b*). The $r$ used to impute the SD of the MD, along with the specification of whether they were estimated or not, are reported in the 'Results' section.

The summary MD of the studies was determined using a random-effects model and reported as mean and 95% confidence (CI) and prediction (PI) intervals (*Higgins, Thompson & Spiegelhalter, 2009*). The CI provides the range in which the true mean effect size is expected to lie; whereas, the PI, reflects the range in which the effect size of a future study is likely to fall (*Higgins, Thompson & Spiegelhalter, 2009*).

Meta-regression analyses were performed to assess the effect of potentially relevant moderators on the MD (*Douglas et al., 2015*; *Song et al., 2001*). Subgroup analysis was not performed for categorical moderators because the selected moderators (*e.g.*, MLSS steady state criteria, CP employed model, CP TTE range) did not report at least three MDs per subgroup (*Douglas et al., 2015*), whereas the meta-regression analyses were performed using meta-regressions for continuous moderators with at least 10 MDs (*i.e.,* CP and

MLSS PO) per model (*Higgins et al., 2023*). For each meta-regression the bubble plots were performed to assess the distribution of the effect size (*Viechtbauer, 2010*).

For each analysis, the heterogeneity was measured using Cochran's test for chi-squared statistic of total (Q) and expected variance and expressed as between study SD ($\tau$).

Additionally, the mean and SD of the MD were used to calculate the LoAs of the Bland–Altman meta-analyses, using the estimation methods recommended by *Tipton & Shuster (2017)* and adapting the R script published by *Ferri Marini et al. (2022b)*. Briefly, both the within-study and between-study variation were used to account for the different sources of variability and compute population pooled LoAs and their outer 95% CIs, which were computed to express the estimated LoA uncertainty.

Moreover, to allow the comparisons of the differences between CP and MLSS across the measurement units (*i.e.,* PO, $\dot{V}O_2$, HR, BLC) used to assess exercise intensity and express the CP and MLSS differences in a common standardized scale (*Dunlap, Dietz & Cortina, 1997*), the standardized mean differences (SMDs) between CP and MLSS were also computed from the MDs and the SDs of the MD separately for each measurement unit. The SMDs expressed in PO and $\dot{V}O_2$ were computed using the build in formula SMCC of the function escalc of metafor (version 3.8.1) (*Viechtbauer, 2010*), which is recommended for paired data points, as it accounts for the presence of data deriving from the same individual by computing SMD from MD and SD of individuals' MD. Specifically, the SMDs for PO and $\dot{V}O_2$ were computed solely for the studies (*Caen et al., 2022*; *Keir et al., 2015*; *Okuno et al., 2011*; *Ozkaya et al., 2022*) presenting both measurement units. Subsequently, the summary SMD and 95% CI and PI were determined for PO and $\dot{V}O_2$ using random-effects meta-analyses (*Viechtbauer, 2010*). Likewise, a sensitivity analysis was performed on the MDs of the studies reporting paired PO and $\dot{V}O_2$ values. The SMDs and sensitivity on the MD were not computed for HR and BLC due to the limited number of studies (*i.e.,* 2) reporting HR or BLC and PO or $\dot{V}O_2$ values.

## RESULTS

### Studies and participants characteristics

All literature search records were examined by title and abstract to exclude the studies that did not meet the eligibility criteria. As reported in Fig. 1 a total of 782 studies were identified based on title and abstract. Overall, 337 studies were downloaded, and full text screened, and finally 10 studies matched the inclusion criteria and were included in the meta-analysis. Most of the studies included in the present article were performed on a cycle ergometer, excluding *Puga et al. (2009)*, where a treadmill was used.

Reported CP values were analyzed independent of the model used for parameter estimation, due to an insufficient number of studies for model comparison. The CP models and MLSS determination criteria are reported in Table 2. In all the studies included in the present article, if insufficient data were reported (*i.e.,* the minimal information needed to compute the paired effect size expressed in PO, HR, $\dot{V}O_2$, and BLC), the authors were contacted by the corresponding author (CFM) *via* email to provide additional information about the included studies.

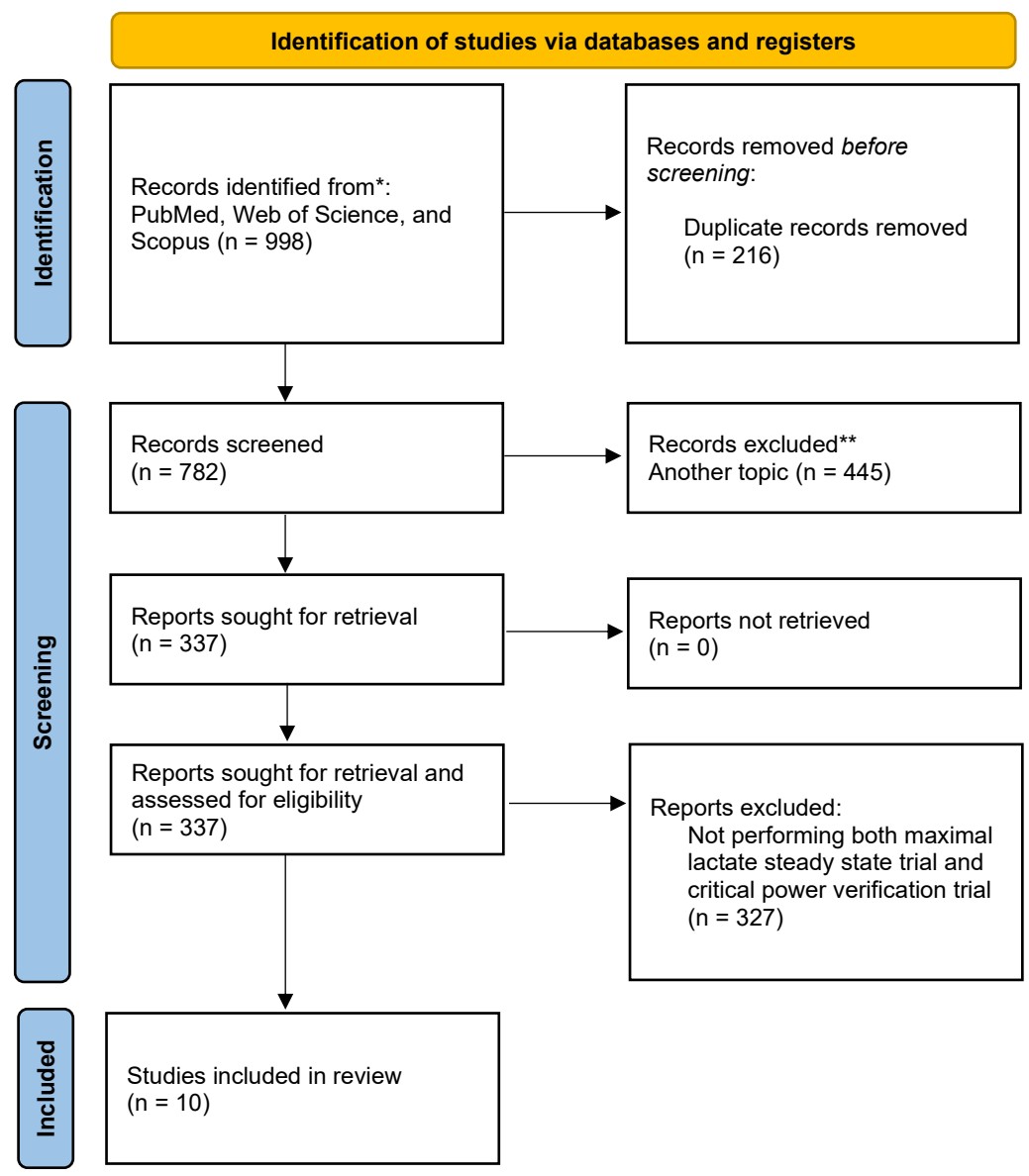

**Figure 1 PRISMA flow chart for new systematic reviews for study selection process.** Adapted from *Page et al. (2021)*. PRISMA, Preferred Reporting Items for Systematic Reviews and Meta-Analyses.

Overall, 111 participants were included in the data analysis; see Table 2 for participants' descriptive data from each study.

The exercise intensities at CP and MLSS, along with their MD and $r$, of the included studies are shown in Table 3.

## Quality evaluation

The quality of the studies included in the review was evaluated by two researchers using the revised Cochrane Risk of Bias (RoB) tool for cross-over trials (*Ding et al., 2015*) (Table 4).

Micheli et al. (2025), *PeerJ*, DOI 10.7717/peerj.19060

**Table 2  Descriptive characteristics of the participants and methodologies of the included studies.**

| Authors | N (f) | Age | Participants | $\dot{V}O_{2max}$ (L·min$^{-1}$) | $\dot{V}O_{2max}$ (mL·kg$^{-1}$·min$^{-1}$) | MLSS (min) | CP model | CP trials TTE (min) |
|---|---|---|---|---|---|---|---|---|
| *Maturana et al. (2016)* | 13 (4) | 26 ± 3 | Healthy young | 4.17 ± 0.68 | 60.4 ± 5.9 | 10–30 | 2-hyp | 1 to 20 |
| *Keir et al. (2015)* | 12 (0) | 27 ± 3 | Healthy young | 4.13 ± 0.52 | – | 10–30 | 3-hyp | 1.5 to 20 |
| *Ozkaya et al. (2022)*[*] | 10 (0) | 21.5 ± 3.4 | Well-trained cyclist | – | 65.4 ± 4.35 | 10–30 | best fit | 2 to 10 |
| *Caritá, Greco & Denadai (2009)* | 6 (0) | 25.5 ± 4.4 | Cyclist | – | 62.7 ± 5.6 | 10–30 | 2-hyp | – |
| *Okuno et al. (2011)*[*] | 10 (0) | 24.4 ± 3.7 | College students | 3.12 ± 0.37 | – | 10–30 | 2-hyp | 2 to 15 |
| *Puga et al. (2009)*[*] | 6 (0) | 23.2 ± 2.7 | Physically active | – | – | 10–30 | – | 3 to 15 |
| *Lievens et al. (2021)* | 12 (0) | 26.1 ± 2.6 | Physical education students | 3.88 ± 0.37 | 51.7 ± 5.9 | 10–30 | best fit | 2 to 20 |
| *Iannetta et al. (2019)* | 11 (5) | 28 ± 7 | Recreationally trained | 3.35 ± 0.68 | – | 15-30 | 3-hyp | 1.5 to 20 |
| *Iannetta et al. (2022)* | 10 (0) | 28 ± 8 | Recreationally cyclists | – | 54.8 ± 6.9 | 10–30 | best fit | 2 to 15 |
| *Caen et al. (2022)a*[*] | 10 (10) | 27 ± 3 | Healthy individuals | 2.62 ± 0.57 | 43.2 ± 7.3 | 10–30 | best fit | 2 to 20 |
| *Caen et al. (2022)b*[*] | 11 (0) | 25 ± 4 | Healthy individuals | 3.69 ± 0.51 | 47.7 ± 5.9 | 10–30 | best fit | 2 to 20 |

**Notes.**

N, total number of participants; f, females; $\dot{V}O_{2max}$, maximal oxygen uptake; MLSS, maximal lactate steady state; CP, critical power; TTE, time to exhaustion; 2-hyp, two parameters hyperbolic model; 3-hyp, three parameters hyperbolic model.

[*]Additional data were provided by the authors.

Micheli et al. (2025), PeerJ, DOI 10.7717/peerj.19060

**Table 3 Descriptive results of the exercise intensities at CP and MLSS of the included studies.**

| Authors | W or %[†] | | | | L·min⁻¹ | | | | bpm | | | | mM | | | |
|---|---|---|---|---|---|---|---|---|---|---|---|---|---|---|---|---|
| | CP | MLSS | MD | r | CP | MLSS | MD | r | CP | MLSS | MD | r | CP | MLSS | MD | r |
| *Maturana et al. (2016)* | 253 ± 44 | 233 ± 41 | 20.0 ± **12.4** | **0.96** | – | 3.54 ± 0.63 | – | – | – | – | – | – | – | 4.4 ± 1.5 | – | – |
| *Keir et al. (2015)* | 226 ± 45 | 223 ± 39 | 2 ± 12 | 0.97 | 3.29 ± 0.48 | 3.27 ± 0.44 | 0.02 ± **0.18** | **0.93** | 162 ± 10 | 161 ± 10 | 1.0 ± **9.7** | **0.53** | – | 6.3 ± 1.4 | – | – |
| *Ozkaya et al. (2022)** | 300 ± 39 | 270 ± 39 | 30.0 ± 10.9 | 0.96 | 4.2 ± 0.5 | 3.9 ± 0.4 | 0.3 ± 0.19 | 0.93 | – | – | – | – | 8.2 ± 1.3 | 5.1 ± 0.5 | 3.1 ± 1.1 | 0.57 |
| *Caritá, Greco & Denadai (2009)* | 314 ± 32 | 287 ± 38 | 26.5 ± 7.4 | 0.99 | 3.88 ± 0.39 | – | – | – | – | – | – | – | 8.4 ± 2.8 | – | – | – |
| *Okuno et al. (2011)** | 267 ± 45 | 254 ± 39 | 12.6 ± 21.3 | 0.88 | 2.52 ± 0.52 | 2.41 ± 0.32 | 0.11 ± 0.55 | 0.21 | 156 ± 8 | 152 ± 10 | 4.6 ± 9.5 | 0.46 | 6.9 ± 2.6 | 5.1 ± 0.9 | 1.8 ± 2.2 | 0.58 |
| *Puga et al. (2009)** | 15.4 ± 1.1[†] | 14.2 ± 1.4[†] | 1.3 ± 1.9[†] | – | 2.95 ± 0.48 | 2.94 ± 0.38 | 0.01 ± 0.21 | 0.90 | 176 ± 4 | 178 ± 5 | −1.9 ± 4.1 | 0.60 | 6.3 ± 1.3 | 6.5 ± 2.0 | −0.2 ± 1.4 | 0.69 |
| *Lievens et al. (2021)* | 250 ± 29 | 241 ± 31 | 9.0 ± 11.0 | 0.93 | – | – | – | – | – | – | – | – | – | – | – | – |
| *Iannetta et al. (2019)* | 214 ± 59 | 215 ± 55 | 1.0 ± **16.6** | **0.96** | – | 2.90 ± 0.67 | – | – | – | – | – | – | – | 5.6 ± 2.0 | – | – |
| *Iannetta et al. (2022)* | 250 ± 47 | 235 ± 46 | 15.0 ± **13.2** | **0.96** | – | 3.46 ± 0.57 | – | – | – | – | – | – | 6.2 ± 1.6 | – | – | – |
| *Caen et al. (2022)a** | 174 ± 39 | 171 ± 41 | 3.0 ± 6.0 | 0.99 | 2.40 ± 0.46 | 2.34 ± 0.49 | 0.06 ± 0.10 | 0.98 | – | – | – | – | – | – | – | – |
| *Caen et al. (2022)b** | 245 ± 37 | 239 ± 38 | 6.0 ± 14.1 | 0.93 | 3.31 ± 0.33 | 3.24 ± 0.34 | 0.07 ± 0.12 | 0.94 | – | – | – | – | – | – | – | – |

**Notes.**

MLSS, maximal lactate steady state; CP, critical power; r, Pearson's correlation coefficient between CP and MLSS; MD, mean differences between CP and MLSS.

*Additional data not available in the full-text were provided by the authors.

[†]Measurement units expressed in % of treadmill.

Bold and italic values indicate data estimated from the median r deriving from the same measurement unit.

Micheli et al. (2025), *PeerJ*, DOI 10.7717/peerj.19060

**Table 4  Quality assessment of the included studies.**

| Item | Maturana et al., 2016 | Keir et al., 2015 | Ozkaya et al., 2022 | Caritá, Greco & Denadai, 2009 | Okuno et al., 2011 | Puga et al., 2009 | Lievens et al., 2021 | Iannetta et al., 2019 | Iannetta et al., 2022 | Caen et al., 2022 |
|---|---|---|---|---|---|---|---|---|---|---|
| 1 | low | low | low | low | low | low | low | low | low | low |
| 2 | high | high | unclear | unclear | high | high | high | high | unclear | high |
| 3 | unclear | unclear | unclear | unclear | unclear | unclear | unclear | unclear | unclear | unclear |
| 4 | low | low | low | low | low | low | low | low | low | low |
| 5 | high | high | high | high | high | high | high | high | high | high |
| 6 | low | unclear | unclear | unclear | unclear | unclear | unclear | low | low | unclear |
| 7 | unclear | unclear | unclear | unclear | unclear | low | unclear | unclear | unclear | unclear |
| 8 | low | low | low | low | low | low | low | low | low | low |
| 9 | low | low | low | low | low | low | low | low | low | low |

**Items:**

1, Appropriate cross-over design; 2, Randomized order of receiving treatment; 3, Carry-over effects; 4, Unbiased data; 5, Allocation concealment; 6, Blinding; 7, Incomplete outcome data; 8, Selective outcome reporting; 9, Other bias.

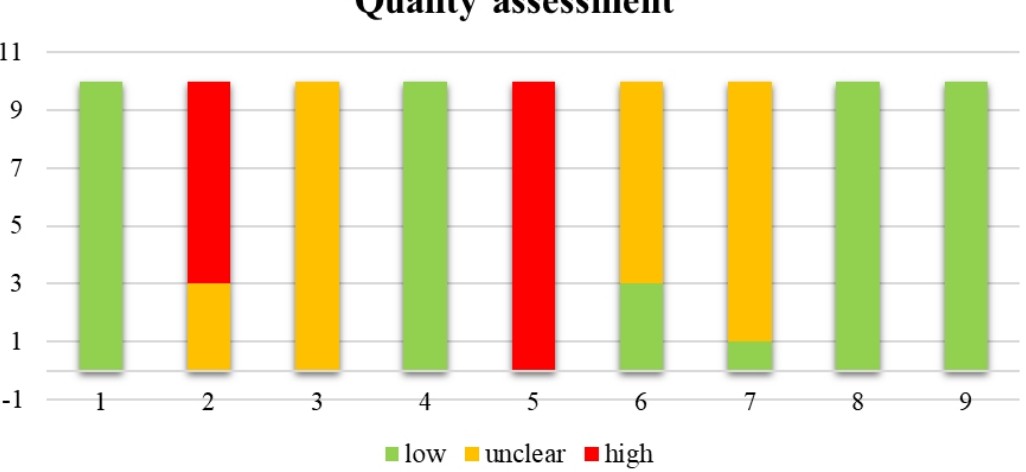

## Quality assessment

**Figure 2 Summary of the quality assessment of the 10 cross-over trials.** 1, Appropriate cross-over design; 2, randomized order of receiving treatment; 3, carry-over effects; 4, unbiased data; 5, allocation concealment; 6, blinding; 7, incomplete outcome data; 8, selective outcome reporting; 9, other bias.

Disagreements between authors were resolved through discussion. The summary of each item considered in study quality assessment are graphically represented in Fig. 2.

### Meta-analyses of the mean differences

The pooled MD in PO between CP and MLSS was 12.42 W (95% CI [4.69–20.16], SE = 3.418, $t = 3.635$, PI = $-11.62$ to 36.47), these differences indicate that, on average, CP is significantly higher ($p = 0.005$) than MLSS. Refer to the forest plot (Fig. 3) for a visual representation of both the pooled MD and CI, and non-pooled MD and CI for each study included in the meta-analysis. The MD expressed in W showed a significant heterogeneity ($Q_{(9)} = 96.490$, $\tau = 10.065$ W, $p < 0.001$). As shown in the SDC2_MD_4_studies, the sensitivity analysis performed on the MDs of the studies reporting paired PO and $\dot{V}O_2$ values showed a pooled MD of 10.59 W (95% CI [$-4.34$–25.51], SE = 5.375, $t = 1.969$, PI = $-24.15$ to 45.32).

In terms of $\dot{V}O_2$, the pooled MD between CP and MLSS was 0.09 L·min$^{-1}$ (95% CI [$-0.02$–0.21], SE = 0.045, $t = 2.036$, PI = $-0.17$ to 0.35). These differences indicate that, on average, CP and MLSS are not different ($p = 0.097$) when expressed as $\dot{V}O_2$. The forest plot in Fig. 4 provides a visual representation of both the pooled MD and CI, and non-pooled MD and CI for each study included in the meta-analysis. The MD, expressed in L·min$^{-1}$, showed a significant heterogeneity ($Q_{(5)} = 15.238$, $\tau = 0.090$ L·min$^{-1}$, $p = 0.009$). As shown in the SDC2_MD_4_studies, the sensitivity analysis performed on the MDs of the studies reporting paired PO and $\dot{V}O_2$ values showed a pooled MD of 0.11 L·min$^{-1}$ (95% CI [$-0.04$–0.25], SE = 0.052, $t = 2.017$, PI = $-0.21$ to 0.42).

The pooled MD in HR between CP and MLSS was 0.61 bpm (95% CI [$-7.70$–8.91], SE = 1.931, $t = 0.313$, PI = $-12.37$ to 13.58), these differences indicate that, on average, CP and MLSS are not different ($p = 0.784$) in terms of bpm. The pooled MD, expressed

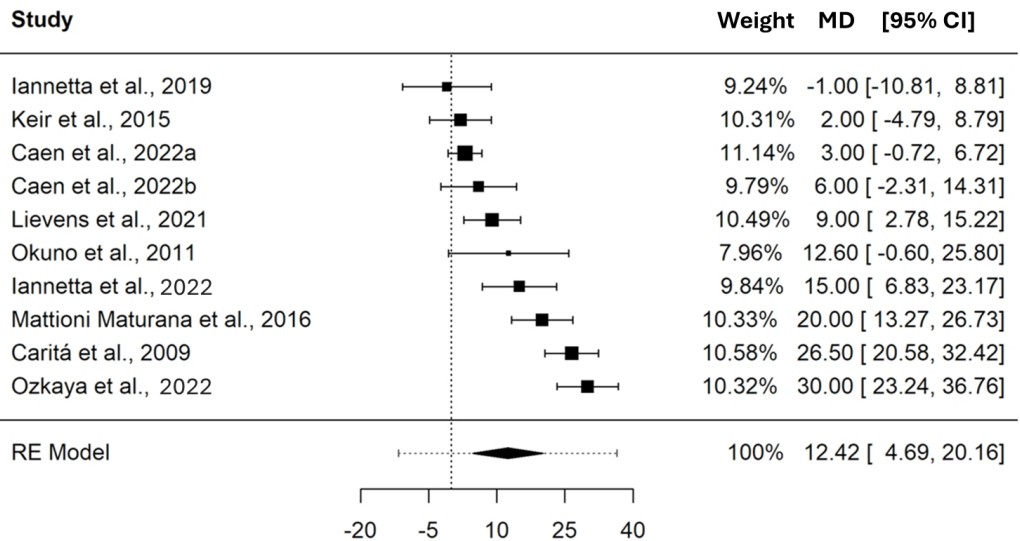

**Figure 3** Forest plot reporting mean difference (MD) and 95% confidence intervals (CI) computed as the difference in power output (W) between critical power and maximal lactate steady state. Studies: *Iannetta et al., 2019*; *Keir et al., 2015*; *Caen et al., 2022*; *Lievens et al., 2021*; *Okuno et al., 2011*; *Iannetta et al., 2022*; *Maturana et al., 2016*; *Caritá, Greco & Denadai, 2009*; *Ozkaya et al., 2022*.

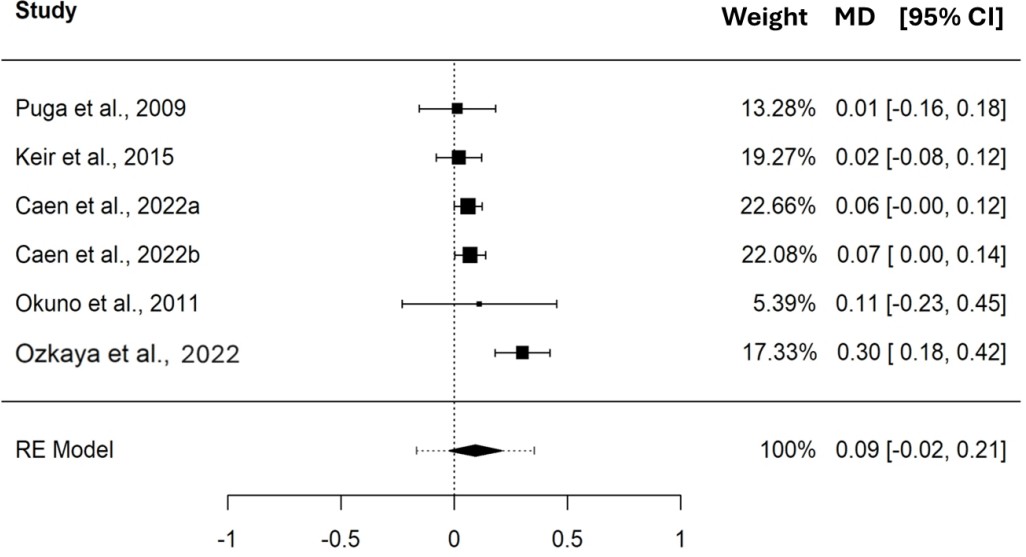

**Figure 4** Forest plot reporting mean difference (MD) and 95% confidence interval (CI) computed as the difference in $\dot{V}O_2$ (L·min$^{-1}$) between critical power and maximal lactate steady state. Studies: *Puga et al., 2009*; *Keir et al., 2015*; *Caen et al., 2022*; *Okuno et al., 2011*; *Ozkaya et al., 2022*.

in bpm, showed a non significant heterogeneity ($Q_{(2)} = 3.767$, $\tau = 2.317$ bpm, $p = 0.152$); see Fig. 5.

The pooled MD in BLC between CP and MLSS was 1.63 mM (95% CI [−2.61–5.86], SE = 0.984, $t = 1.655$, PI = −6.50 to 9.75), these differences indicate that, on average,

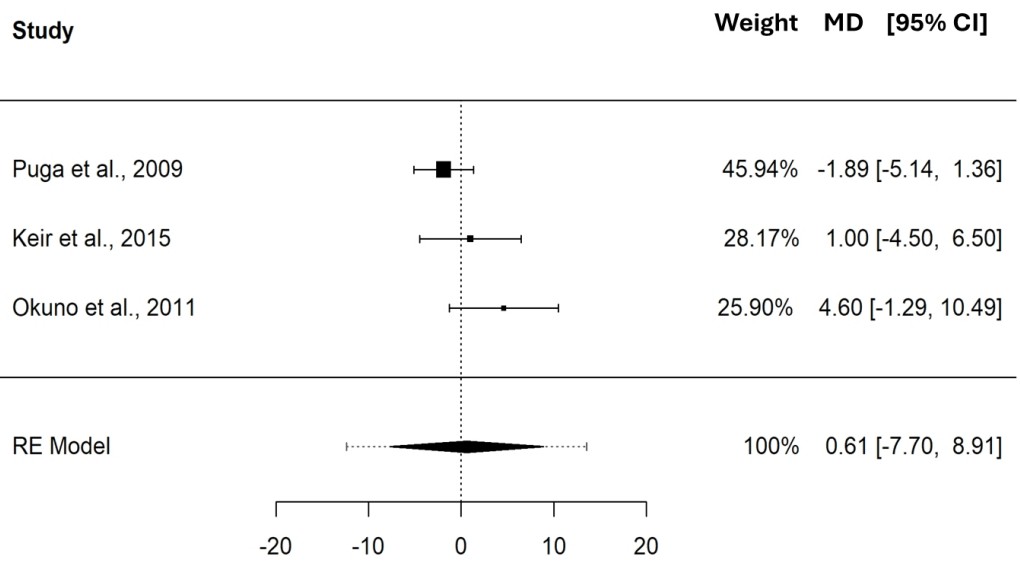

**Figure 5** Forest plot reporting mean difference (MD) and 95% confidence interval (CI) computed as the difference in HR (bpm) between critical power and maximal lactate steady state. Studies: *Puga et al., 2009*; *Keir et al., 2015*; *Okuno et al., 2011*.

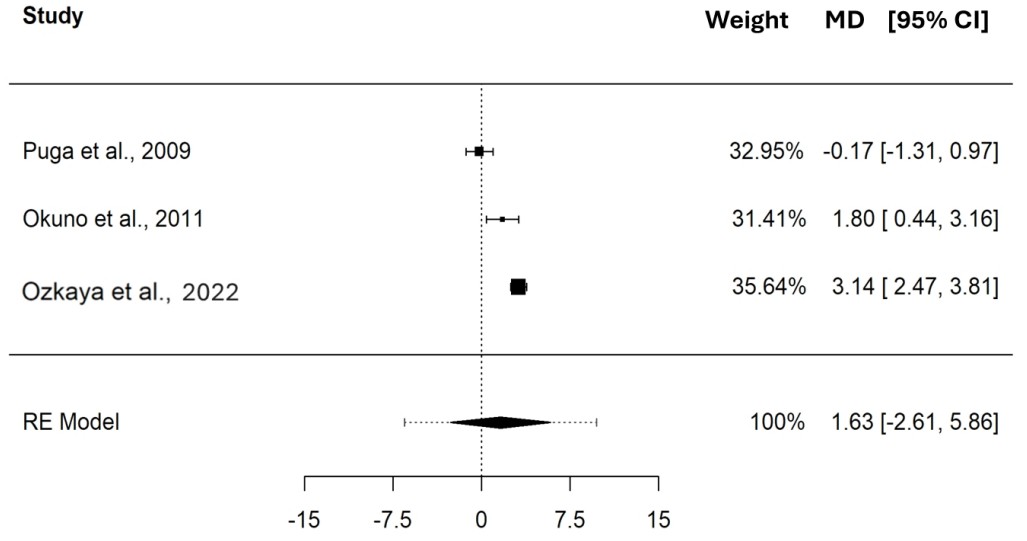

**Figure 6** Forest plot reporting mean difference (MD) and 95% confidence interval (CI) computed as the difference in BLC (mM) between critical power and maximal lactate steady state. Studies: *Puga et al., 2009*; *Keir et al., 2015*; *Okuno et al., 2011*.

CP and MLSS are not different ($p = 0.240$) in terms of mM. Please refer to the forest plot (Fig. 6) for a visual representation of both the pooled MD and CI, and non-pooled MD and CI for each study included in the meta-analysis. The MD, expressed in mM, showed significant heterogeneity ($Q_{(2)} = 24.352$, $\tau = 1.612$ mM, $p < 0.001$).

**Table 5   Meta-regressions' estimates of the effects of critical power (CP) and maximal lactate steady state (MLSS) intensities on the mean differences between CP and MLSS expressed in W.**

| Moderators | IV | B | SE of B | t | p | CI$_{INF}$ of B | CI$_{SUP}$ of B | τ |
|---|---|---|---|---|---|---|---|---|
| **MLSS (W)** | | | | | | | | |
| | Slope | 0.243 | 0.072 | 3.396 | 0.009 | 0.078 | 0.408 | |
| | Intercept | −44.634 | 17.031 | −2.621 | 0.031 | −83.907 | −5.361 | 6.389 |
| **CP (W)** | | | | | | | | |
| | Slope | 0.21 | 0.045 | 4.673 | 0.002 | 0.107 | 0.314 | |
| | Intercept | −39.378 | 11.306 | −3.483 | 0.008 | −65.45 | −13.306 | 4.796 |

**Notes.**

IV, independent variable; B, unstandardized beta coefficient; SE, standard error; *t*, *t*-value of regression coefficient *t*-test; *p*, probability value associated with *t*; CI, inferior (INF) and superior (SUP) 95% confidence intervals of B; Intercept and slope, coefficients of the meta-regressions; τ, estimated standard deviation of underlying effects across studies.

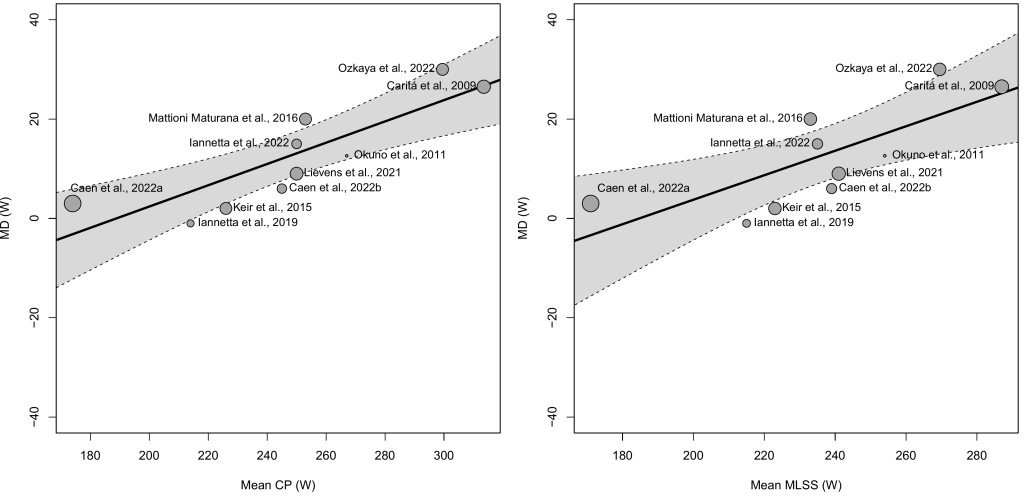

**Figure 7   Bubble plots showing the association between critical power (CP) and maximal lactate steady state (MLSS) mean differences (MD) and the mean CP (A) or MLSS (B) in W.** The size of the points is drawn proportional to the weight that the studies received in the analysis (with larger points for studies that received more weight).

## Meta-regressions analysis on the mean difference

The meta-regression analyses showed significant heterogeneities on the MD expressed in W for the meta-analysis having CP ($Q_{(8)} = 22.426$, $p = 0.004$) and MLSS ($Q_{(8)} = 32.868$, $p < 0.001$), as moderators, with a τ equal to 4.796 W and 6.389 W, respectively. Test of moderators results for CP ($F_{(1,8)} = 21.091$, $p = 0.002$) and MLSS ($F_{(1,8)} = 11.114$, $p = 0.010$), revealed a significant effect on the MD expressed in W (see Table 5 for the estimated effects of the moderator using meta-regressions and Fig. 7 for a graphical representation by means of bubble plots).

Meta-regression analyses on the MDs expressed in $\dot{V}O_2$, HR, and BLC were not performed since these variables presented less than 10 MDs per model (*Higgins et al., 2023*).

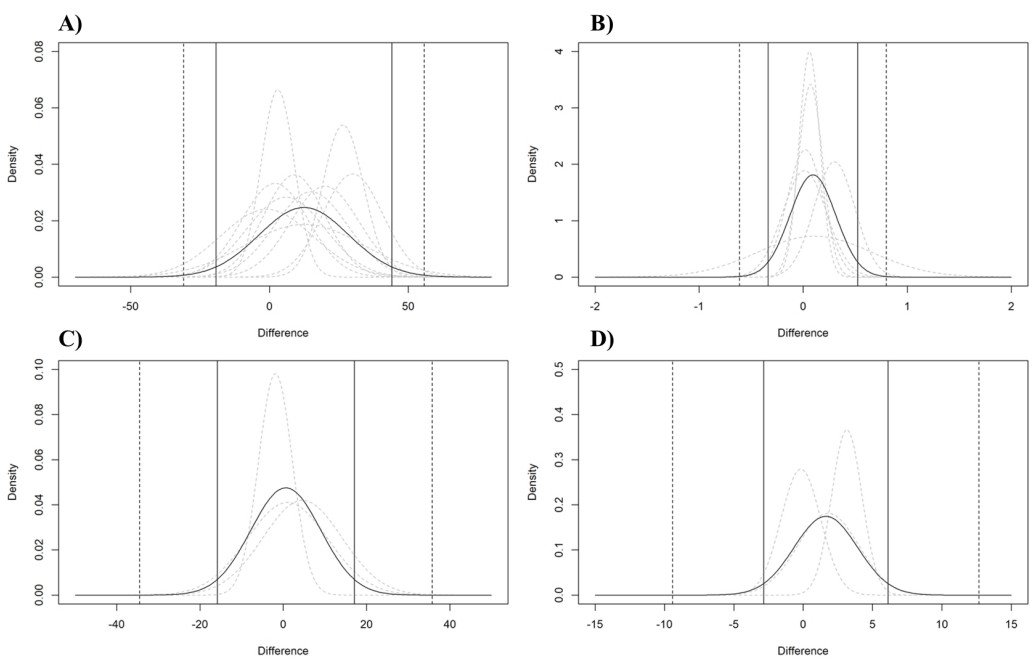

**Figure 8 Analysis of the agreement between CP and MLSS.** Pooled (black curve) and individual studies' (grey dashed curve) estimated normal distributions of the differences between MLSS and CP expressed as W (A), L·min$^{-1}$ (B), bpm (C), and mM (D). The vertical lines indicate the estimated pooled limits of agreements (solid vertical lines) and their outer 95% confidence intervals (dashed vertical lines).

## Analysis of agreement

The analysis of the agreement between CP and MLSS showed a mean bias of 12.42 W (95% LoA = −19.23 to 44.08, outer 95% CI of LoA [−30.91–55.76]), 0.09 L·min$^{-1}$ (95% LoA = −0.34 to 0.52, outer 95% CI of LoA 95% [−0.61–0.80]), 0.61 bpm (95% LoA = −15.84 to 17.05, outer 95% CI of LoA 95% [−34.53–35.74]), and 1.63 mM (95% LoA = −2.85 to 6.11, outer 95% CI of LoA 95% [−9.41–12.67]) for PO, $\dot{V}O_2$, HR, and BLC, respectively; see Fig. 8.

## Meta-analyses and comparisons of the standardized mean differences expressed in W and L·min$^{-1}$

The pooled SMD between CP and MLSS in PO and $\dot{V}O_2$ were 0.648 (95% CI [−0.189–1.485], PI = −1.118 to 2.414, $\tau$ = 0.560, SE = 0.301, t = 2.150, p = 0.098) and 0.482 (95% CI [−0.044–1.007], PI = −0.402 to 1.365, $\tau$ = 0.256, SE = 0.189, t = 2.543, p = 0.064), respectively. The results of the SMD expressed in PO and $\dot{V}O_2$ for each study are reported in the SDC3_SMD.

## DISCUSSION

The present meta-analysis represents the first attempt to understand how the utilization of different exercise intensity indicators (*i.e.*, PO, $\dot{V}O_2$, HR, and BLC) affect the proximity and agreement between CP and MLSS, by analyzing the MD, the SMD, and the agreement

between the two indices. Although recent meta-analyses (*Borszcz et al., 2024*; *Galan-Rioja et al., 2020*) investigated the associations between these indices (or their surrogates), by analyzing the correlations between indexes (*Galan-Rioja et al., 2020*) and the effect of methodological factors (*Borszcz et al., 2024*), both studies did not assess how the use of different exercise intensity indicators affects the agreement between CP and MLSS. Our main finding indicates that CP is significantly higher than MLSS by an average of 12.42 W in terms of PO. However, no significant differences were observed when CP and MLSS were compared based on $\dot{V}O_2$, HR, and BLC. Higher PO at CP than MLSS was expected and aligns with previous research suggesting that CP tends to be higher than MLSS (*Dekerle et al., 2003*; *Maturana et al., 2016*; *Nixon et al., 2021*; *Pringle & Jones, 2002*). The lack of significant difference in $\dot{V}O_2$ values between CP and MLSS is also consistent with previous studies (*Caen et al., 2022*; *Keir et al., 2015*). The lack of a significant difference between CP and MLSS in terms of $\dot{V}O_2$, HR, and BLC could also be because studies that provided the internal load data reported among the smallest differences between CP and MLSS in terms of PO. Indeed, due to the lack of data, it was not possible to assess the differences between CP and MLSS in terms of other exercise intensity indicators (*e.g.,* $\dot{V}O_2$, HR, and BLC) besides PO in three out of the four studies where the largest differences in PO were observed (*Caritá, Greco & Denadai, 2009*; *Iannetta et al., 2022*; *Maturana et al., 2016*). Therefore, to minimize the impact of incomplete data and between studies differences on the different exercise intensity indicators, a sensitivity analysis was conducted using only studies reporting multiple exercise intensity indicators to more robustly compare the responses of these indicators in CP and MLSS. In the present study, due to incomplete data reported in several studies (see Table 3), a comparison across all exercise intensity indicators (*i.e.,* PO, $\dot{V}O_2$, HR, and BLC) was not possible. Indeed, only one study (*Ozkaya et al., 2022*) provided upon request all the data needed for the calculation of the MD for all exercise intensity indicators. Likewise, the comparison between exercise intensity indicators was only possible for W and L·min$^{-1}$ (reported in four studies with five ES), since bpm and mM data were reported in just two studies. Nonetheless, the sensitivity analysis conducted between the exercise intensity thresholds for the most reported exercise intensity indicators (*i.e.,* $\dot{V}O_2$ and PO, representing internal and external load, respectively), showed consistent results when considering studies reporting paired PO and $\dot{V}O_2$ values. Indeed, this analysis revealed that the pooled MDs for the five ESs (MD: 10.59 W; 0.11 L·min$^{-1}$) were similar to those of the overall dataset (MD: 12.42 W; 0.09 L·min$^{-1}$) for both PO and $\dot{V}O_2$ (see SDC2_MD_4_studies).

Furthermore, when paired W and L·min$^{-1}$ ESs were rescaled in a standardized measurement unit (*i.e.,* SMD computed as CP minus MLSS), the difference between CP and MLSS tended to be higher when expressed in PO (0.648) compared to $\dot{V}O_2$ (0.482). However, it is necessary to interpret this result with caution, since two of the five SMD expressed in $\dot{V}O_2$ were higher than the SMD expressed in PO (see SDC3_SMD). The heterogenous results found among studies further suggest that differences in the SMD expressed in PO and $\dot{V}O_2$ could be a result of the protocol used to estimate CP and MLSS. This aspect regarding the concordance between CP and MLSS in terms of PO was recently highlighted by a study by *Caen et al. (2024)*, who concluded that the discrepancies between

CP and MLSS can be reconciled if very stringent criteria are used in their determination procedures.

Additionally, the findings derived from the analysis of the moderators (*i.e.,* fitness status expressed as POs at CP and MLSS) showed that the difference in PO between CP and MLSS tends to increase in fitter individuals (*i.e.,* participants with higher CP and MLSS in terms of PO). These results could be attributed to the methodology used to determine the CP and MLSS. Indeed, CP determination requires exhaustive testing, whereas assessing MLSS does not. It is widely acknowledged that aptitude and familiarity with maximal exertion could influence test scores by contributing to the achievement of actual exhaustion during testing procedures. Therefore, it can be hypothesized that individuals who have higher levels of fitness or greater experience with maximal efforts may achieve higher values of CP while the MLSS, whose test does not require maximal efforts, remains unaltered, thus leading to a larger difference between CP and MLSS. Importantly, the results of the present meta-analyses should be carefully extrapolated and contextualized since the homogeneity of the sample considered. Indeed, the studies included in the present work were all on young and fit individuals, whereas no studies were conducted on older adults and clinical populations. Moreover, the present work highlighted the presence of a sex bias in the studies, which enrolled only 19 females out of 111 participants. Thus, research on the relationship between CP and MLSS should include comprehensive studies on different populations, examining potential differences related to participants' sex, age, and fitness level to ensure the external validity in MMSS research.

In the present study, the agreement analysis between CP and MLSS showed wide LoAs for PO (*i.e.,* $-19.23$ to $44.08$ W), $\dot{V}O_2$ (*i.e.,* $-0.34$ to $0.52$ L·min$^{-1}$), HR (*i.e.,* $-15.84$ to $17.05$ bpm), and BLC (*i.e.,* $-2.85$ to $6.11$ mM), suggesting that, regardless of a small mean bias, there is a confirmation of a great individual variability when comparing CP and MLSS.

The importance of this aspect should not be underestimated as it suggests that, considering the high interindividual variability, in some individuals, the proximity between CP and MLSS may be remarkably lower than the mean difference reported in the present study, due to the low accuracy and precision of measurements. This could potentially result in significant errors at an individual level, with implications for exercise prescription from a practical perspective.

The results of the present study confirm that CP is consistently higher than MLSS and are in line with those found from *Maturana et al. (2016)*, in which only one out of 13 participants achieved a stable BLC while cycling at the PO derived from CP. The authors also noted the potential influence of data modeling strategies and chosen TTEs range on deriving accurate estimates of CP, highlighting the need for more rigorous methods and guidelines for an appropriate CP estimation (*Maturana et al., 2016*).

However, it is imperative to move beyond whether CP overestimates MLSS and focus on understanding and resolving the methodological shortcomings that lead to discrepancies between CP and MLSS. A recent study conducted by *Iannetta et al. (2022)* suggests that variations in estimation methods can contribute to modify the discrepancy between CP and MLSS. Indeed, the authors discovered that the differences in PO between CP and MLSS can be reconciled if specific testing strategies are used. In particular, the discrepancies vary

based on the models used for CP estimation and the criteria to establish MLSS (*i.e.,* the time interval in which there must be a steady state in BLC), with an improved concordance between CP and MLSS when MLSS is estimated with narrower temporal criteria (*i.e.,* steady state between 15th or 20th and 30th min) and it is essential that future research accurately measure CP and MLSS. Indeed, previous studies have demonstrated that the accuracy of the CP prediction can be improved if a power-time relationship is constructed meticulously by performing the appropriate number of exhaustive trials (*i.e.,* three to five, reaching a standard error (SE) in CP lower than 5%) and including proper TTE range (*i.e.,* shortest trial 2–3 min long and the longest more than 10 but no longer than 15 min) (*Jones et al., 2019*; *Muniz-Pumares et al., 2019*). On the other hand, if the sensitivity of the MLSS test is increased with small differences in POs or speeds between trials, from a practical standpoint, any disparities between CP and MLSS can be eliminated (*Iannetta et al., 2022*). Furthermore, a recent longitudinal study of *Caen et al. (2022)* provided additional evidence supporting the need for more rigorous methods for determining CP and MLSS, suggesting that discrepancies between CP and MLSS can be attributed to differences in the methodological approaches utilized. Indeed, in almost all their participants, CP and MLSS were found to occur in very close proximity, with only three out of 42 comparisons exhibiting a discrepancy greater than 10 W. Therefore, based on the results of *Caen et al. (2022)* and those of the present study, it is reasonable to expect minor and practically irrelevant differences between CP and MLSS if more rigorous methodological approaches for CP and MLSS determination are used.

Understanding factors that contribute to this variability may offer valuable insights for developing targeted interventions or guidelines on accurate CP and MLSS estimation. In this regard, the above-mentioned aspects are in line with the findings of *Borszcz et al. (2024)*, who argued that, because of the methodology in CP and MLSS determination, caution should be used when directly comparing these two indices in terms of PO. Additionally, although exercise intensity can be prescribed and monitor using different exercise intensity indicators, such as HR and BLC, no studies assessed how the use of different indicators affect the CP and MLSS relation.

The results of the present study should, however, be considered with caution because the inclusion of studies employing heterogeneous methodologies. For example, the studies by *Keir et al. (2015)* and *Iannetta et al. (2019)* utilized the 3-parameter model, which may have influenced the comparisons, as highlighted by *De Lucas (2018)*. Methodological considerations were consistently taken into account in this systematic review; for instance, the study by *Okuno et al. (2011)* was included, despite its use of intermittent tests, because the availability of paired (MLSS and CP) data points was deemed to mitigate this potential bias. In the study by *Iannetta et al. (2022)*, which provided data based on different methodologies, the CP with the smallest error (*i.e.,* best fit) and the MLSS using the most commonly used approach for its determination (*i.e.,* one mmol/L increase between minute 10 and 30) were selected. Caution is also advised when interpreting the results expressed in HR and BLC as only three studies were included in this analysis for these variables.

Additionally, a possible limitation of the present study is the use of only data deriving from prolonged trials at CP and MLSS which had limited the number of studies included for the PO when an estimated CP was available. However, this choice was made in order to allow a more stringent comparison between different indicators (*i.e.,* PO, $\dot{V}O_2$, HR, and BLC) including only prolonged trials.

Furthermore, a limitation of the present systematic review is its inability to assess the effect of the moderators and their impact on the exercise intensity indicators ($\dot{V}O_2$, HR, BLC) other than PO due to the limited number of studies reporting the exercise intensity at CP and MLSS. Specifically, the influence of methodological moderators on the methods used to assess CP (*e.g.*, number of trials to exhaustion, range of time to exhaustion, and mathematical model used for its determination) and MLSS (*e.g.*, criteria for determining the presence or absence of a steady state in the BLC, PO intervals (*i.e.*, delta PO) used between tests) could not be analyzed due to the scarcity of studies reporting the aforementioned information and the heterogeneity of the methods. Thus, it is crucial that future studies provide a more detailed overall description of the physiological responses to allow a better understanding of the exercise intensity profiles based upon different exercise intensity indicators and that, as suggested by *Mesquida et al. (2022)*, more data sharing be encouraged to improve the overall quality of meta-analyses in sports science.

The lack of standardized procedures for determining CP and MLSS remains a major issue that contributes to the heterogeneity of results. Therefore, until consistent methodologies are established, prudence is essential when trying to synthesize findings from studies employing different methodologies, as highlighted by *Borszcz et al. (2024)*.

The findings of this study may also extend to the real-world setting. Indeed, the present study showed that, not only CP was higher than MLSS in terms of PO but, due to the large interindividual variability in the differences between CP and MLSS, the accuracy of threshold-based exercise prescription could yield a high error in a large portion of population if CP and MLSS are used interchangeably.

## CONCLUSIONS

The results of the present systematic review show that, when expressed as PO, CP tends to result greater if compared to MLSS and their difference is affected by both the age and fitness status of the individuals, with younger and fitter subjects having greater discrepancies between CP and MLSS. However, there were no differences between CP and MLSS in terms of $\dot{V}O_2$, HR, and BLC.

Additionally, the high interindividual variability in the differences between CP and MLSS points out the possible high error in predicting one parameter from the other at an individual level. Finally, the present systematic review confirms the need for more standardized and robust testing procedures for CP and MLSS. Therefore, future studies addressing the methodological issues associated with each index proposed to identify the MMSS and their agreement, and guarantee standardized testing procedures, are crucial before determining the superiority of one index over the other (if any) as well as their possible interchangeability.

## ACKNOWLEDGEMENTS

We would like to thank Professor Jan Boone, Doctor Kevin Caen, Professor Nilo Massaru Okuno, Professor Gleber Pereira, Professor Ozgur Ozkaya, and Professor Guilherme Morais Puga for providing us additional and unpublished data. Additionally, we would like to thank Andrea Del Bianco, librarian at the University of Urbino Carlo Bo, for the support provided during the literature search.

### Funding

This work has been funded by the European Union - NextGenerationEU under the Italian Ministry of University and Research (MUR) National Innovation Ecosystem grant ECS00000041 - VITALITY - CUP H33C22000430006. The funders had no role in study design, data collection and analysis, decision to publish, or preparation of the manuscript.

### Grant Disclosures

The following grant information was disclosed by the authors:
European Union - NextGenerationEU under the Italian Ministry of University and Research (MUR) National Innovation Ecosystem: ECS00000041.
VITALITY: CUP H33C22000430006.

### Competing Interests

Carlo Ferri Marini is an Academic Editor for PeerJ.

### Author Contributions

- Lorenzo Micheli conceived and designed the experiments, performed the experiments, analyzed the data, prepared figures and/or tables, authored or reviewed drafts of the article, and approved the final draft.
- Francesco Lucertini conceived and designed the experiments, authored or reviewed drafts of the article, and approved the final draft.
- Tommaso Grossi performed the experiments, analyzed the data, prepared figures and/or tables, and approved the final draft.
- Silvia Pogliaghi conceived and designed the experiments, authored or reviewed drafts of the article, and approved the final draft.
- Daniel A. Keir conceived and designed the experiments, authored or reviewed drafts of the article, and approved the final draft.
- Carlo Ferri Marini conceived and designed the experiments, performed the experiments, analyzed the data, prepared figures and/or tables, authored or reviewed drafts of the article, and approved the final draft.

### Data Availability

  This is a systematic review/meta-analysis.

## Supplemental Information

Supplemental information for this article can be found online at http://dx.doi.org/10.7717/peerj.19060#supplemental-information.

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
