# Peer review of "Analysis of the factors influencing the proximity and agreement between critical power and maximal lactate steady state: a systematic review and meta-analyses"

_PeerJ, doi:10.7717/peerj.19060_

## Round 0.1 · original submission · Major Revisions

Dear authors,

Please revise the manuscript considering the reviewers suggestions.

Thank you.

Best regards.

Reviewer 1 ·

Basic reporting

Lines 110-112: Key-words “respiratory compensation point” and “thresholds” are not listed in ‘SDC1_Keyword_search_strategy.pdf’ please delete.

Please use ‘et al.’ in all citations of studies in figures and tables.

Some paragraphs are excessively long. For example, paragraphs spanning lines 36 to 73 and 328 to 364 should be divided into smaller units.

Experimental design

Lines 152-154: Please add the correlation coefficients used for impute the SD of MD of each outcome, also is important to report in what studies it was imputed.

Lines 164-166: In my opinion, and the opinion of some important statisticians, the I² statistic has limitations. I suggest reporting tau values (which are intrinsically reported as prediction intervals) and not I². Please see:
a) https://www.sciencedirect.com/science/article/pii/S1836955320300163
b) https://www.bmj.com/content/342/bmj.d549.full
c) https://onlinelibrary.wiley.com/doi/full/10.1002/jrsm.1230

Lines 244-250 and Table 5: Please include the tau values from the meta-regressions. Tau values are highly relevant information, as they can be compared to the tau value presented in the meta-analysis of PO. Lower tau values from the meta-regression compared to the meta-analysis value indicate that the predictors in the meta-analysis accounted for some of the heterogeneity between the studies. The Pseudo-R² statistic can provide further insight into this. Please see:
a) https://www.metafor-project.org/doku.php/faq#for_mixed-effects_models_how_i
b) Konstantopoulos, S and Hedges, LV. Statistically analyzing effect sizes: Fixed and random-effects models. In Cooper H, Hedges LV, Valentine JC Editors. The handbook of research synthesis and meta-analysis. New York: Russell Sage Foundation, 2019. pp. 245–280
c) Borszcz, F. K., De Aguiar, R. A., Costa, V. P., Denadai, B. S. & De Lucas, R. D. 2024. Agreement Between Maximal Lactate Steady State and Critical Power in Different Sports: A Systematic Review and Bayesian´s Meta-Regression. J Strength Cond Res, 38, e320-e339.

Validity of the findings

I have concerns about the inclusion of some effect sizes from certain studies. Moreover, I believe it is problematic to conduct a “simple” meta-analysis that arrives at a meta-analyzed mean when the effect sizes come from MLSS and CP tests that were determined using highly heterogeneous methodologies (see Borszcz et al. JSCR, 2024 findings). The question that remains is, what does this meta-analyzed mean represent?

To address some of these concerns, the authors conducted meta-regressions including PO in MLSS, PO in CP, and age of participants as predictors. However, the results of the meta-regressions do not advance knowledge or overcome the limitations of the "simple" meta-analysis because PO in MLSS and CP are again influenced by the methodology applied in these two tests. Regarding age, the range is so short that "significant" coefficients may simply be residuals of other predictors not included in the meta-regression. In my opinion, to solve the problems between MLSS and CP, the methodological characteristics of these tests should be included as predictors in the meta-regression.

Regarding the authors' findings, they concluded that MLSS and CP are significantly different only in relation to PO, but not in relation to VO2, HR, and BLC. It is important to note that not all studies reported all outcomes. Therefore, a valid comparison of whether MLSS and CP differ in relation to PO and VO2 (PO and HR, PO and BLC) should only include studies that report both outcomes.

Results: At no point is it stated that the Puga et al. (2009) study was conducted on running exercise, while all other studies were conducted on cycling.

Results: Please discuss, Okuno et al (2011) inclusion. That study used an intermittent protocol for MLSS and CP assessments, which involved 30 seconds of exercise followed by 30 seconds of rest. This approach typically leads to higher power outputs for both MLSS and CP compared to continuous protocols. For instance, their intermittent CP was 267 ± 45 W, while the "continuous" CP in the same study was 151 ± 30 W. Notably, all other studies in the meta-analysis used continuous exercise protocols.

Table 2. It is unclear why the authors chose to include Iannetta 2019, which defines MLSS with a 1 mmol/L BLC change between the 15th and 30th minutes. In contrast, they focused solely on the MLSS 10-30 criterion from Iannetta 2022, although this study presented multiple MLSS criteria.

Table 3. Please include the MD ± SD of MD for all included studies. Given that you performed a meta-analysis, you have calculated these effects. Additionally, for any imputed effects, please specify that what SD of the MD was imputed.

Table 3. I am unclear about the decision to include the following studies: Maturana, 2016; Caritá, 2009; Lievens, 2021; Iannetta, 2019, 2022. While they conducted constant load trials at MLSS and CP, they did not report VO2, HR, or BLC for one or both conditions.

Lines 244-250: Please add bubble plots (https://www.metafor-project.org/doku.php/plots:meta_analytic_scatterplot) to show the relationship between the predictors and MD of each meta-regression.

Lines 257-262: As the authors previously stated in the statistical analysis that SD of MD were imputed, it is unclear how this imputation might have affected the precision of the limits of agreement (LoA) estimates. Imputations are common in meta-analysis for estimating standard errors of effect sizes (i.e., the precision or weight of the study). However, in such cases, the SD is the point estimate used to derive the LoA. In this regard, Galan-Rioja et al. (2020) reported that correlation coefficients between MLSS and CP varied between -0.11 and 0.99, which could potentially cause imprecisions due to the limited number of r values reported by the authors in Table 3

Lines 282-289: The authors stated: “Additionally, the lack of a significant difference between CP and MLSS when the internal load (i.e., VO2, HR, and BLC) was considered could also be since the studies that reported the internal load data reported among the smallest differences between CP and MLSS in terms of PO. In contrast, in three (Caritá et al., 2009, Iannetta et al., 2022, Maturana et al., 2016) out of the four studies where the largest differences were found in terms of PO, more analysis than that of PO was not possible due to a lack of data. Thus, it is crucial that future studies provide a more detailed overall description of the internal load responses to allow a better understanding of the exercise intensity profiles based upon different parameters.” Regarding this, a recent meta-regression by Borszcz et al. (2024) demonstrated that differences in PO between MLSS and CP are primarily attributed to methodological variations in these tests. Since the authors did not address these methodological issues, including studies with diverse methodologies would result in inappropriate conclusions. I recommend the author conduct a subgroup analysis, encompassing only studies that reported both PO and VO2, PO and HR, and PO and BLC, to derive a more precise conclusion on this matter. Therefore, conclusions about the interchangeability of PO-VO2, PO-HR, and PO-BLC could be drawn based on the studies that derived both outcomes.

Reviewer 2 ·

Basic reporting

The authors aimed to explore the agreement between critical power and MLSS, in terms of power output, VO2, HR and blood lactate responses, throughout a systematic review and meta-analysis.
Although it is generally well presented (and written), there are several concerns in this research, which imply in the (non)acceptance for publication.

Experimental design

This topic has been extensively explored in two recent SRMA (Galan-Rioja et al, 2020 and Borszcz et al. 2024). Although, authors recognize (and cite) these studies, it seems that they avoid reporting the findings. One of aspects that authors emphasise as a novelty in the current SRMA, is the analysis of VO2, HR and blood lactate relative to the critical power and MLSS (not solely power output). However, it is not clear how these physiological variables were extracted, to attribute to the intensities of CP and MLSS.
Another concern is about the missing studies in the SR. There are at least six potential studies that could/should be included (based on the criteria of inclusion) e.g. Pallares et al. (2020), Fontana et al. (2017), Pringle and Jones (2002), Greco et al. (2012), Dekerle et al. (2003), Sperlich et al. (2014). In this sense, it is not clear enough the criteria for excluding some studies in the Figure 1. E.g. what means "no useful data". In addition, why incremental test was a criterion for include the studies? Concluding, I am not sure that you have included all of studies that focused on the comparison aimed in the analysis. Please, read with attention the study of Borszcz et al. (2024), who have systematically explored the studies and the covariables associated to the determination of both indices, as well the participants characteristics.
There is no mention that the study is analysing only cycling data. I recommend to authors including this information throughout the whole manuscript.

Validity of the findings

Authors concluded that the power output present an important mean difference between critical power and MLSS, but in terms of VO2, HR or blood lactate does not. I would challenge the authors to explain how it is possible. If the external load (i.e. power output) determine the metabolic load, how could I cycle with a power e.g. 15W higher and have the same VO2 (based on the general efficiency of 10mlO2/min/W)? It seems that this outcome is a result of how these physio data were extracted (e.g. from incremental or square-wave testing). In addition, this conclusion was based on six studies (i.e VO2) where there are mixture of critical power models, and no mention how the VO2 was extracted to refers to both indices.

Additional comments

Specific comments:

Abstract:
Lines 3-4. About this statement, it is not established in your study.

Introduction

Lines 50-51 and 61-66: The definition and explanation about the MLSS criteria has been extensively studied by Beneke and colleagues in the last 30 years. These sentences should be refined in light of those studies.

Lines 92-95: The main aim of study is not pioneering. The secondary objective seems to be more ‘innovative’.

Methods
Lin 124: Why including Portuguese language, and not Spanish or German, for instance?

Line 131- 135: This topic need more information. The authors should describe which authors were contact and what data was provided etc...
How did you extract the physiological data? i mean, e.g. VO2-CP was extrapolated from the incremental test? And about VO2-MLSS? Was it assessed during a continuous exercise, as normally done in MLSS trials? Please refine this information. It is crucial for interpreting the results.

Lines 155-156: This information is not clear, please refine what you mean for prediction interval.

Results
Table 2: The studies of Keir and Iannetta, 2019, have used the 3-parameter model, and it biased the comparison between the indices (Table 3). Please check the studies of Bull et al. and the recent SRMA of Borscz et al. (2024).

Discussion

Line 268: Are you sure that this is the first attempt to compare CP and MLSS?
Line 272: And what about the Borszcz et al. (2024) study? It seems that the authors overlooked this study.
Line 273-275: It makes no sense. Please provide a strong rationale for explaining this result.
Line 279: The study of Keir et al (2015) is biased by the 3-parameter model employed for modelling the critical power, as highlighted in de Lucas (2018). It should be considered/discussed in the present analysis.
Line 287: Did you ask those data to the authors?
Line 306: Why could it potentially influence on the differences?
Line 312: This was tested and reported in the study of Borczsz et al. (2024). Again, it seems that this study is overlooked.
Line 337-338: Are you sure that Iannetta et al. discovered the differences reported in this sentence?
Line 347: Jones et al here is an intriguing citation. This group of authors have already used a 10km performance in elite runners for (wrongly) modelling CS (exceeding the 15min proposed). In addition, Caen et al. (2022) have used a range of 2-20 min for determining critical power. Not surprising, in this study the difference between VO2-CP and VO2-MLSS was narrow.

Conclusion
Line 384: This is not a novelty.
Line 386: This study does not analyse how the indices represent the boundary between the cited domains. Instead, the study only check how the indices agree (or not).

---

## Round 0.2 · Minor Revisions

Dear Authors,

Both reviewers suggested improvements regarding the last manuscript version. Please revise the document considering all the details indicated.

Thank you.

Best regards.

Reviewer 1 ·

Basic reporting

Line 179: Please add “between study SD” before the tau symbol.

SDC_3 different font sizes were used in the table title, please revise.

Line 250: Please add W after the tau value, it is expressed in the same unit of measurement as MD. Also, at line 262 add: L/min, line 272: bpm, line 281: mM, and line 288: W after tau values.

Experimental design

Lines 205-210: It is unclear how you computed the SMD. Did you first calculate the SMD between CP and MLSS for PO, then the SMD between CP and MLSS for VO2, and finally compared these SMDs (in this point the correlation was applied?) getting a “net” SMD? Please specify which SD was used to standardize the difference (MLSS SD, CP SD, MLSS and CP SD pooled, or SD of the differences) (please see: https://onlinelibrary.wiley.com/doi/full/10.1002/sim.10114).

Briefly, did you use the following equations to derive the SMDnet (net effect)?:
SMDnet = SMD[PO] – SMD[VO2]
SEnet = √(SE[PO]² + SE[VO2]^2 – 2 x r x SE[PO] x SE[VO2])

It is unclear why the SMD point estimate changed with different correlation values. I would expect only the standard errors to change.


Lines 205-210: In my view, by using a correlation of zero, you assumed that there is no relationship between the differences in MLSS and CP for VO2 and the differences for PO. However, it is well-established that a higher PO produces a higher VO2, and they are linearly related. Therefore, I suggest assuming a correlation of 0.50, which is sufficiently conservative.

Validity of the findings

No corrections were necessary concerning the validity of the findings for this review phase. I would like to thank the authors for addressing all of my previous comments.

Additional comments

Lines 338-339: Revise the use of “trend towards significance”. There is no definition of a trend toward statistical significance in the literature, and, therefore, describing “nearly significant” results as a trend introduces substantial subjectivity and the opportunity for biased reporting language that could mislead a reader.

Lines 450-453: I simply want to emphasize that this is a very important point.

Reviewer 2 ·

Basic reporting

Congratulations to the authors for the revised version, which was really improved.

Experimental design

In regard of the language of articles (during the search), I guess the authors should amend the sentence, as they reported English, Italian and Portuguese, not solely Portuguese aside of English.

Validity of the findings

There are still one critical point I consider necessary for improving the quality of the results and interpretation of the current SRMA. I remain unconvinced by the response regarding the exclusion of some potential studies. The authors stated: “We did not include the cited studies because they lacked a verification trial for critical power (CP) (Pallares et al. (2020), Pringle and Jones (2002), Dekerle et al. (2003)], or it was unclear how certain data (e.g., VO₂ at CP in Greco, 2012) was retrieved”. I understand this justification for excluding these studies from the VO₂ analysis. However, it does not apply to the W analysis. Given that these studies presented a comparison of power output between CP and MLSS, the meta-analysis without this data should be considered biased.

Additional comments

About Sperlich et al. (2014), the full reference is that:

Sperlich B, Zinner C, Trenk D, Holmberg HC. Does a 3-minute all-out test provide suitable measures of exercise intensity at themaximal lactate steady state or peak oxygen uptake for well-trained runners? Int J Sports Physiol Perform 9: 805–810, 2014.

BUT, as they using running exercise and the 3-min all-out test, I understand its exclusion.

---

## Round 0.3 · Minor Revisions

Dear Authors,

Please revise the manuscript considering the minor suggestions by reviewer 1.

Thank you.

Best regards.

Reviewer 1 ·

Basic reporting

No comments.

Experimental design

Line 198: “escal” is “escalc”, please correct.

Lines 200-201: The formula described in the paper was not meaningful. It reports the delta SMD between PO and VO2 divided by the pooled SD of PO and VO2, which are derived from different units of measurement (i.e., watts and L/min). A more appropriate approach would be to calculate the SMD as such in a controlled pre-post-study design:
1. Calculate the pre-post SMD for the control group.
2. Calculate the pre-post SMD for the experimental group.
3. Compare these two SMDs (i.e., calculate the delta SMD).

In your study, you should calculate the SMD CP-MLSS for PO and the SMD CP-MLSS for VO2, and then compare these two SMDs." Briefly, you should use the following equations to derive the ∆SMD:

1 – Power output
SMD[PO] = (MLSS[PO] – CP[PO])/(√(SD MLSS[PO]² + SD CP[PO]²)/2)
SE[PO] = √(SD MLSS[PO]² + SD CP[PO]² – 2 x r x SD MLSS[PO] x SD CP[PO])

2 – VO2
SMD[VO2] = (MLSS[VO2] – CP[VO2])/(√(SD MLSS[VO2]² + SD CP[VO2]²)/2)
SE[PO] = √(SD MLSS[VO2]² + SD CP[VO2]² – 2 x r x SD MLSS[VO2] x SD CP[VO2])

3 – delta SMD
Then, the SMDs and standard errors (SE[PO] and SE[VO2]) of stages 1 and 2 are used to calculate the delta SMD:
∆SMD = SMD[PO] – SMD[VO2]
∆SE = √(SE[PO]² + SE[VO2]² – 2 x r x SE[PO] x SE[VO2])

Validity of the findings

No comments.

Additional comments

No comments.

Reviewer 2 ·

Basic reporting

No comment.

Experimental design

No comment.

Validity of the findings

No comment.

Additional comments

The quality of paper improved enough for publication.

---

## Round 0.4 · accepted · Accept

Dear Authors,

Thank you for addressing all of the reviewers' comments during the review process.

Best regards.

Reviewer 1 ·

Basic reporting

None.

Experimental design

None.

Validity of the findings

None.

Additional comments

None.